



# Water limitation may restrict the positive effect of higher temperatures on weathering rates in forest soils

Salim Belyazid[1], Cecilia Akselsson[2], Giuliana Zanchi[2]

[1] Department of Physical Geography, Stockholm University, Stockholm, SE-223 62, Sweden

[2] Department of Physical Geography and Ecosystem Science, Lund University, Lund, SE-223 62, Sweden

Corresponding author: Salim Belyazid (salim.belyazid@natgeo.su.se)



# 1 Abstract

Climate change is generally expected to have a positive effect on weathering rates, due to the strong temperature dependence of the weathering process. There are, however, a number of feedback effects, both positive and negative, that can affect the weathering response to climate change, but that have not been fully taken into account in previous estimates. Important feedback mechanisms are the direct effect of changes in soil moisture, and the indirect effects through tree growth and decomposition on weathering rates. In this study, the dynamic forest ecosystem model ForSAFE, with mechanistic descriptions of tree growth, decomposition, weathering, hydrology and ion exchange processes, is used to investigate the effects of future climate scenarios on weathering rates in a more holistic way than has been done before. 544 productive coniferous forest sites, part of the Swedish National Forest Inventory, are modelled, and differences in weathering responses to changes in climate from two Global Climate Models are investigated. The study shows that weathering rates are likely to increase, but not to the extent predicted by a direct response to elevated air temperatures. The simulations show that increases in soil temperatures are less evident than those in air temperature, thereby dampening the effect of warming on weathering. Most importantly, the study shows that soil moisture availability has a strong potential to limit the expected response to increased temperature. While changes in annual precipitation may not indicate further risk for more severe water deficits, seasonal differences show a clear difference between winters and summers. Taking into account the seasonal variation, the study shows that reduced soil water availability in the summer seasons will strongly limit the expected gain in weathering associated with higher temperatures.





## 2  Introduction

The use of the natural environment for economic activity has led to increasingly evident environmental
impacts. On the global scale, these impacts are manifested through the exceedance of the planetary
boundaries (Steffen et al., 2015) including climate change (IPCC, 2014). Awareness about these impacts is
driving a shift away from conventional, linear resource use to a circular, bio-based economy (Bennich and
Belyazid, 2017). Sweden is engaging on an ambitious national strategy to have no net emission of
greenhouse gases by the year 2045 (Regeringskansliet, 2017), supported by a national strategy for a bio-
based economy (FORMAS, 2012). While forestry is already today a key sector in this strategy (Lundmark
et al., 2014), further increase in biomass production from forestry is needed to meet the stated goals
(Börjesson et al., 2017, Cintas et al., 2017).

Higher levels of biomass harvesting from forests may, however, compromise the sustainability of forest
ecosystems (de Jong et al., 2017). As in other parts of Europe, Swedish forests have been exposed to acid
atmospheric deposition (Ref), the effects of which can still be seen in acidified forest soils (Iwald et al,
2013; Akselsson et al., 2013; Pihl Karlsson et al., 2011). Forest soil acidification may be further exacerbated
by the removal of alkaline cations though biomass harvesting (Iwald et al., 2013; Belyazid et al., 2006;
Akselsson and Belyazid, 2018; Zetterberg et al., 2013). Iwald et al. (2013) show that the net extraction of
base cations through forest harvesting in spruce forests causes higher soil acidification than the current
acid deposition. Using production forecasts from Claesson (2008) and Iwald et al. (2013) conclude that
forestry will be the major source of soil acidification in the coming 100 years. These findings are supported
by mass balance studies by Akselsson et al. (2007, 2016) demonstrating net long-term losses of
exchangeable base cations under different harvesting intensities, even under climate change scenarios
leading to higher weathering rates.

Base cation release through mineral weathering sustains the long-term supply of base cations (Akselsson
et al., 2016; Aherne et al., 2012). In light of increasing temperatures following climate change, this
compensation potential is anticipated to increase as weathering is responsive to temperature changes
(Oelkers and Schott, 2001; Sverdrup and Warfvinge, 1993; Eiriksdottir et al., 2013). Akselsson et al. (2016)
used the PROFILE model, which includes an Arrhenius dependence of weathering rates on temperature,
to show that weathering rates would increase by 20% to 33% in Swedish soils in response to the expected
warming by between 2.2°C and 3.3°C over the coming 50 years. While this increase falls short of the
expected increase in losses through whole tree harvesting at 66% as compared to conventional stem
harvesting (Aklselsson et al., 2016), it is in line with the independent finding of Aherne et al. (2012) using
a different method in comparable ecosystems.

Besides its response to temperature, mineral weathering is regulated through a range of other factors,
many of which involve geochemical, physical as well as biological processes (Erlandsson et al., 2016;
Sverdrup and Warfvinge, 1993; Palandri and Kharaka, 2004; Oelkers et al., 1994). The potential
acceleration of weathering rates in response to higher temperatures can be cancelled by drier soil
conditions, which in turn can be driven by increased plant transpiration following higher plant growth. At
the same time, higher plant growth can produce more litterfall, thereby increasing the concentrations of
organic radicals, lowering base cation concentrations through uptake, and lowering pH, thus further
promoting weathering. When considering the different pathways through which climatic changes can
affect weathering, it becomes less evident whether the net effect will be positive or negative.





To investigate the net response of weathering to climatic changes, it is necessary to account for different processes simultaneously. For this, integrated ecosystem models are well suited tools. In this study we use the integrated forest ecosystem model ForSAFE (Wallman et al., 2004; Belyazid et al., 2006; Zanchi et al., 2014) to evaluate the processes by which climate change affects mineral weathering in forest soils, and identify instances where these processes amplify or cancel out each other. The study focuses specifically on the effect of higher temperatures and different precipitation patterns on mineral weathering rates in the unsaturated zone. It will test the hypothesis that a reduction in water availability will limit the expected increase in weathering expected from higher temperatures. To account for the uncertainty of future climate trajectories, modelled data from two climate models with different projections are used.

## 3   Method

### 3.1   The ForSAFE model

ForSAFE is an integrated forest ecosystem model that simulates the interlinked biogeochemical cycles of water, carbon, nutrients and other elements (Belyazid, 2006; Zanchi et al., 2014; Belyazid et al., 2017, Yu et al., 2016). Base cation release through weathering in ForSAFE is based on the concept of the PROFILE model (Sverdrup and Warfvinge, 1993), where four dissolution rates are parameterised for each mineral. The dissolution rates in ForSAFE are driven by soil moisture, H+ concentrations in the soil solution, partial soil solution CO2 pressure, organic anions in the soils solution (Warfvinge and Sverdrup, 1992; Sverdrup and Warfvinge, 1993). All dissolution rates are regulated by temperature responses according to the Arrhenius equation, while only the H+ and H2O dissolution rates are subject to retardation functions in response to elevated base cation and Al concentration in the aqueous phase (Sverdrup and Warfvinge, 1993).

To calculate the rates of release of the different base cations, the dissolution rates are applied to the different minerals present in the different layers of the simulated soil. Each mineral is associated with a set of kinetic parameters that govern the dissolution rates, and a specific stoichiometry used to translate the mineral dissolution rate to element release rates (Akselsson, 2004; Sverdrup and Warfvinge, 1993).

The weathering module in ForSAFE is integrated with a soil solution chemistry module, that also communicates with a routine for organic matter decomposition, a routine for hydrology, and a routine for photosynthesis, tree growth, allocation, uptake and litter fall (See Belyazid, 2006 or Wallman et al., 2005 for detailed descriptions of the ForSAFE model). Based on the concept by Aber and Federer (1992), photosynthesis is driven by the foliar nitrogen content and the size of the canopy that is able to absorb solar radiation, and constrained by water availability through transpiration. Transpiration and nutrient uptake are driven by the requirements of the trees, and regulated by the availability of water and dissolved nutrients in the soil. The length of the growing season as well as the set-off of photosynthesis are promoted by higher air temperatures, and so is the vapour pressure difference and thereby transpiration. Litter production returns both carbon and nutrients to the soil in the form of litter, and are subsequently released into the soil solution through the decomposition process based on the original concept by Walse et al (1998). The decomposition process responds positively to increases in temperature and moisture, meaning that it is also retarded by lack of water.



In ForSAFE, mineral weathering and the associated release of base cations follows four pathways governed by soil moisture, soil solution pH, partial $CO_2$ pressure in the soil solution and the concentrations of organic radical (in the form of dissolved organic carbon DOC) in the soil solution. All pathways are subject to the Arrhenius temperature response depending on the different activation energies specific to each reaction and mineral as given in Akselsson et al. (2016). Moisture on the other hand has a more complex effect on weathering rates. While increased soil moisture always promotes the dissolution of the different minerals, it also governs element concentrations that have dissimilar consequences on weathering rates. Higher soil moisture also reduces the concentrations of $H^+$, DOC and $CO_2$, thus slowing weathering rates. At the same time, lower concentrations of base cations (particularly $Ca^{2+}$) and aluminium ions alleviate the brakes on weathering. Moisture deficit also slows down plant growth and nutrient uptake, thereby limiting the simulated indirect effects of plants on weathering.

### 3.2   Forest sites data

ForSAFE was applied to 544 managed forest sites in Sweden. Soil data was originally compiled by Warfvinge and Sverdrup (1995) and complemented by Alveteg (2004) with data from the Swedish National Forest Inventory. Unlike the study by Alveteg (2004), only soil data was used as ForSAFE internally calculates tree growth, decomposition and hydrology related fluxes. The modelled soils represent the rooting zones and are contained within a depth of up to 50cm.

ForSAFE requires time series data for atmospheric deposition and climate for the simulation period between the years 1900 and 2100. Data for atmospheric deposition for NH4+, NO3- and SO42- were derived from the EMEP model simulations (Simpsson et al., 2012), adopting the historical trends from Schöpp et al. (2003) and the future projections following the emissions scenario of the current legislation of the revised Gothenburg protocol of the Long Range Transboundary Air Pollution convention (LRTAP). The atmospheric deposition data for the Ca2+, Mg2+, K+, Na+ and Cl- came from recent output by the MATCH model (Langner et al., 1996) and held constant over the simulation period. Forest stand history and future management were derived from current recommendations for spruce and pine forests in the management tables (*gallringsmallar*) developed by the Swedish Forest Agency and indications given by the model INGVAR (Swedish Forest Agency, 1985a, 1985b; Jacobson et al., 2008). According to these sources the rotation length and number of thinnings vary according to the geographical location of the stand and the site productivity. The forest management was further corrected based on information on the stand age from the National Forest Inventory (Belyazid and Zanchi, 2019).

### 3.3   Climate

Precipitation and temperature data for the period between 1961 and 2008 were derived from historical records from the weather stations of the Swedish Meteorological Institute (SMHI), and spatially interpolated to the exact coordinates of the simulated sites (David Rayner, personal communication). Data on global radiation time-series from the NCEP/NCAR's reanalysis project were converted to photosynthetically active radiation (PAR) using SMHI's STRÅNG model (Rayner, personal communication). The sites specific climate data for the period 1961 to 2008 was then used to calibrate the historical and future climate trends from two Global Climate Models (GCM): the the Max Plank Institute's ECHAM5 GCM (hereafter referred to as ECHAM) and the National Center for Atmospheric Research's CCSM3 GCM (hereafter referred to as CCSM). The SRES A2 story line was used for the future climate.

Although using the same emissions scenario, the two climate models produce markedly different projections of temperature and precipitation. ECHAM predicts a stronger increase in winter temperatures



(December to February), while CCSM predicts more markedly warmer summers (June to August) (Figure 1). Both models predict wetter winters, but different geographical patterns in summer precipitation (Figure 1). While ECHAM divides Sweden into an evenly drier south and wetter north during the summers, CCSM foresees wetter inland areas and drier coastal areas, with no apparent north to south pattern (Figure 1).

For the purpose of the study, the forest ecosystem model ForSAFE was run with three climate scenarios: 1- the baseline scenario assumes no further change compared to the reference period of 1981-2010, 2- the CCSM scenario follows the climate projections of the CCSM model, and 3- the ECHAM scenario follows the projections of the ECHAM model. All comparative results are shown as averages for the period 2071-2100.

## 4 Results

### 4.1 Annual weathering rates

The rates of base cation release from weathering vary greatly between sites under the baseline scenario, but with no clear large scale geographical patterns, except for the relatively lower rates in central Sweden due to poorer parent material (Figure 2). The average weathering rate under the base line scenario is 0.35 keq.ha$^{-1}$.yr$^{-1}$, with the median at 0.24 keq.ha$^{-1}$.yr$^{-1}$. The variability is relatively large, with the standard deviation at 0.4 keq.ha$^{-1}$.yr$^{-1}$, and the skewness strongly positive at 4.43. Under future climate change, the average annual weathering rate is expected to increase to 0.43 keq.ha$^{-1}$.yr$^{-1}$ and 0.44 keq.ha$^{-1}$.yr$^{-1}$ using the CCSM and ECHAM climate scenarios respectively. The variability of the weathering rates is expected to be wide with standard deviation equal to 0.48 and 0.49 under CCSM and ECHAM respectively (Table 1). The expected average annual weathering rates in response to climate change do not show any clear geographical patterns corresponding to the expected temperature increase, but reflect the soil mineralogy in central Sweden (Figure 2).

### 4.2 Annual change in weathering following climate change

Weathering rates are expected to increase consistently throughout the country and under both climate change models (Figure 3). The geographical pattern of the increase in weathering mirrors more closely the increase in soil temperature (Figure 3) than the increase in air temperature (Figure 1).

The annual average response of weathering to soil temperature increase is also stronger than that to air temperature increase (Table 2). The average annual weathering rates are expected to increase by 6.6% to 6.7% per °C increase in air temperature under the CCSM and ECHAM scenarios respectively. When compared to the increase in soil temperature, weathering rates increments per degree Celsius rise to 9.3(%·°C$^{-1}$) and 8.9(%·°C$^{-1}$) respectively under CCSM and ECHAM. The difference in weathering response to air temperature as compared to soil temperature change is even more expressed during the winter season, but virtually absent during summers (Table 2).

The increase in weathering correlates positively and significantly to soil temperature increase under both climate scenarios (Figure 4). However, the spread of the correlations remains wide as shown by the low R$^2$ values (Figure 4). According to the linear regression (Figure 4), the incremental increase in weathering with soil temperature increase is 6.0%·°C$^{-1}$ and 8.6 %·°C$^{-1}$ under CCSM and ECHAM respectively.

The effect of the change in annual average soil moisture on the weathering rates seems inconclusive (Figure 4). Under the CCSM climate scenario, the correlation between the increase in average annual





weathering and the decrease in average annual moisture is not significant, weak, and has a wide spread. Under the ECHAM climate scenario, although the correlation is significant, it has a wide spread and a week slope (Figure 4).

### 4.3  Change in weathering during the summer season

Unlike the change in annual soil temperatures shown above, the increase in summer soil temperatures is different between the two climate scenarios (Figure 5). Summer soil temperatures are expected to increase by 3.5°C to 4.5°C according to CCSM, but only by 2.5°C to 3.5°C according to ECHAM. Yet, although summer soil temperatures would increase the most under CCSM, weathering rates do not follow suite as expected. The effect of the increase in summer soil temperatures on weathering is only significant under the ECHAM scenario, with a substantial slope of the linear regression between the increase in weathering rate and that in soil temperature of 8%·°C⁻¹, but still with a wide spread reflected in the low $R^2$ of 0.06. Under CCSM, the correlation between soil temperature change and the change in weathering is not significant.

The change in summer soil moisture is substantial with an average of -7.6% ($\sigma$=1.9%) under CCSM and -5.6% ($\sigma$=1.4) under ECHAM compared to the no climate change scenario. The change in summer soil moisture is stronger and more consistent than the change in summer precipitation, which shows a reduction by an average of -5.9% ($\sigma$=8.2%) under CCSM and an average of 1.1% ($\sigma$=7.8%) under ECHAM.

The change in summer weathering correlates significantly to the change in summer soil moisture, with higher increases in weathering corresponding to lower soil moisture reductions (Figure 6).

### 4.4  Winter

There are clearer regional differences in the change of winter soil temperatures under both scenarios, with a higher increase in the southern half of Sweden (Figure 7). Winter soil temperatures will be up to 2.5°C higher under CCSM as compared to the base scenario, and as high as 4°C under ECHAM (Figure 7). The geographical pattern of winter temperature increase is also more clearly reflected in the change of weathering rates as compared to the annual and summer changes show above (Figures 3 and 5).

Under both scenarios, the change in weathering rates correlates significantly and strongly to the change in soil temperature (Figure 8). On the other hand, the change in winter soil moisture, from -0.5 to +0.5%, indicates no or marginal further water constrain on weathering under the two climate scenarios.

## 5  Discussion

The annual weathering rates calculated by ForSAFE are well within the range presented in Akselsson et al. (2019), which compiles weathering rates calculated by seven different methods on comparable soils in Sweden. The annual average weathering calculated in this study, at 0.35 keq.ha⁻¹.yr⁻¹, is similar to that estimated by Akselsson et al. (2016) at 0.33 keq.ha⁻¹.yr⁻¹ using the steady state model PROFILE. The results of this study confirm the conclusion of Akselsson et al. (2016) about the importance of moisture changes in regulating the response of weathering to increased temperatures. Due to the dynamic nature of the model, moisture is internally simulated in the present study, providing an integrated assessment of the net effect of climate change on weathering.

Both Akselsson et al. (2016) and Kronnäs et al. (2019) show the importance of soil moisture on weathering estimates, implying that part of the disparity can be explained by differences in soil moisture between the



models used to estimate wearthering rates. Yet, as Kronnäs et al. (2019) shows, the difference between the PROFILE and ForSAFE models applied on soils with exactly the same geophysical properties is small. One important difference between Akselsson et al. (2016) and this study is how soil coarse material is treated. The coarse particles (diameter higher than 2mm) are considered inert as their reactive area is negligible compared to that of fine earth. Depending on the estimation method, differences in the fraction

of coarse material can occur. Finally, the user defined possible minerals have a direct and potentially strong effect on weathering estimates, since they form the substrate available for weathering. In the ForSAFE database, limestone is considered a possible mineral, unlike the PROFILE database which excluded it (Akselsson et al., this issue).

The present study is the first to calculate weathering rates using a fully dynamic ecosystem model on a

national scale. The dynamic representation of the weathering, tree growth and decomposition processes in ForSAFE enables modelling of the overall effects of climate change, through direct changes in temperature and precipitation as well as indirect changes related to forest management, on nutrient cycling and acidification in forest soils, taking the feedbacks between the processes into account. Whereas PROFILE only could give a "maximum effect", assuming that only the change in temperature affects

weathering rates (Akselsson et al., 2016), ForSAFE could give a more balanced picture, also taking heat transfer between air and soil, and soil moisture changes, into account.

The temporal resolution in ForSAFE allows analysis of seasonal differences, as demonstrated in Kronnäs et al. (2019). The temporal resolution of ForSAFE means a great advantage compared with steady-state models such as PROFILE. The interannual variation of weathering rates in a future climate, as compared

to the interannual variation of other processes such as nutrient uptake from vegetation and decomposition, is of key importance for predicting effects on nutrient cycling and acidification of forest soils in a future climate.

Reservations remain however about the accuracy of the estimates of weathering rates; as possible sources of uncertainty have not been addressed on the national scale investigated in this study. Casetou-

Gustafson et al. (2018) compared weathering rates where the mineralogy estimations were based on regionally generalised information about occurrence of different minerals, with weathering rates where actual measurements of mineral occurrence were used to estimate mineral contents, on two sites. They concluded that the two approaches gave similar overall weathering rates. However, in-depth analyses of the weathering rates of specific minerals revealed short-comings that could affect the results in cases with

other mineral compositions. Thus, Casetou-Gustafson et al. (2018) argue that site specific properties can reduce the uncertainties in weathering estimations.

The importance of soil moisture for weathering rates was demonstrated in this study, through the hampering effect on the temperature-induced increase in weathering rates during the summer months. The results are in agreements with the conclusions from Kronnäs et al. (2019), where weathering rates

from PROFILE and ForSAFE were compared for two sites. The modelling in that study was based on the same input data for both models, but an important difference was that ForSAFE dynamically modelled moisture, whereas for PROFILE it was an input, based on rough field assessments. Although the resulting weathering rates from the two models were of comparable sizes, there were significant differences especially for one of the sites, which could largely be explained by differences in moisture.

Soil moisture is of key importance also for other processes, such as tree growth as demonstrated in Belyazid and Zanchi (in press), and decomposition (Moore, 1986). Accordingly, Kronnäs et al. (2019)




suggested evaluation of moisture modelled with ForSAFE, on sites with moisture measurements.Except for soil moisture, exposed mineral surface has in several studies been identified as a parameter which greatly influence the overall uncertainties of weathering rates modelled with PROFILE (Jönsson et al.,
Hodson et al., 1996, Zak et al., 1997). The exposed mineral surface area of soils is often calculated using regressions from BET measurements (Warfvinge and Sverdrup, 1995). To reduce the uncertainties connected to those regressions, a revision of the regression is required, where the newest technology can be used.

## 6  Acknowledgements

This work was financed through the research program on Quantifying Weathering Rates for Sustainable Forestry (QWARTS) funded by FORMAS.

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





Table 1: Statistical summary of the release rates of $Ca^{2+}+Mg^{2+}+K^{+}+Na^{+}$ from mineral weathering (called weathering rates) in the unsaturated soil (between 45cm and 50cm depth) at 544 forested sites

| | Annual[1] weathering rate without climate change (keq.ha$^{-1}$.yr$^{-1}$) | Annual[2] weathering rate with future climate according to CCSM (keq.ha$^{-1}$.yr$^{-1}$) | Annual[2] weathering rate with future climate according to ECHAM (keq.ha$^{-1}$.yr$^{-1}$) |
|---|---|---|---|
| Minimum | 0.05 | 0.05 | 0.05 |
| Maximum | 4.42 | 5.61 | 5.70 |
| Mean | 0.35 | 0.43 | 0.44 |
| Median | 0.24 | 0.30 | 0.29 |
| Std Deviation | 0.40 | 0.48 | 0.49 |
| Skewness | 4.43 | 4.66 | 4.65 |

[1]Average annual release of $Ca^{2+}+Mg^{2+}+K^{+}+Na^{+}$ (keq.ha$^{-1}$.yr$^{-1}$) through mineral weathering for the period 2071-2100 under a hypothetical future with no change in climate compared to the period 1981-2010.

[2]Average annual release of $Ca^{2+}+Mg^{2+}+K^{+}+Na^{+}$ (keq.ha$^{-1}$.yr$^{-1}$) through mineral weathering for the period 2071-2100 under a changing climate according to climate forecasts by CCSM and ECHAM.



Table 2: Average (±standard deviation) increment of weathering per degree Celsius increase of air (rows 1 and 3)  and soil temperature (rows 2 and 4) (%·°C$^{-1}$) for 544 productive coniferous forest sites under CCSM and ECHAM.

|  |  | Annual | Winter | Summer |
|---|---|---|---|---|
| CCSM | $\Delta_{Weathering}/\Delta_{AirTemp}$ | 6.7 ±1.6 | 2.6 ±2.1 | 7.3 ±1.6 |
|  | $\Delta_{Weathering}/\Delta_{SoilTemp}$ | 9.3 ±1.8 | 8.9 ±3.3 | 7.5 ±1.6 |
| ECHAM | $\Delta_{Weathering}/\Delta_{AirTemp}$ | 6.6 ±1.5 | 4.3 ±2.3 | 7.9 ±1.7 |
|  | $\Delta_{Weathering}/\Delta_{SoilTemp}$ | 8.9 ±1.3 | 9.4 ±1.8 | 7.7 ±1.6 |





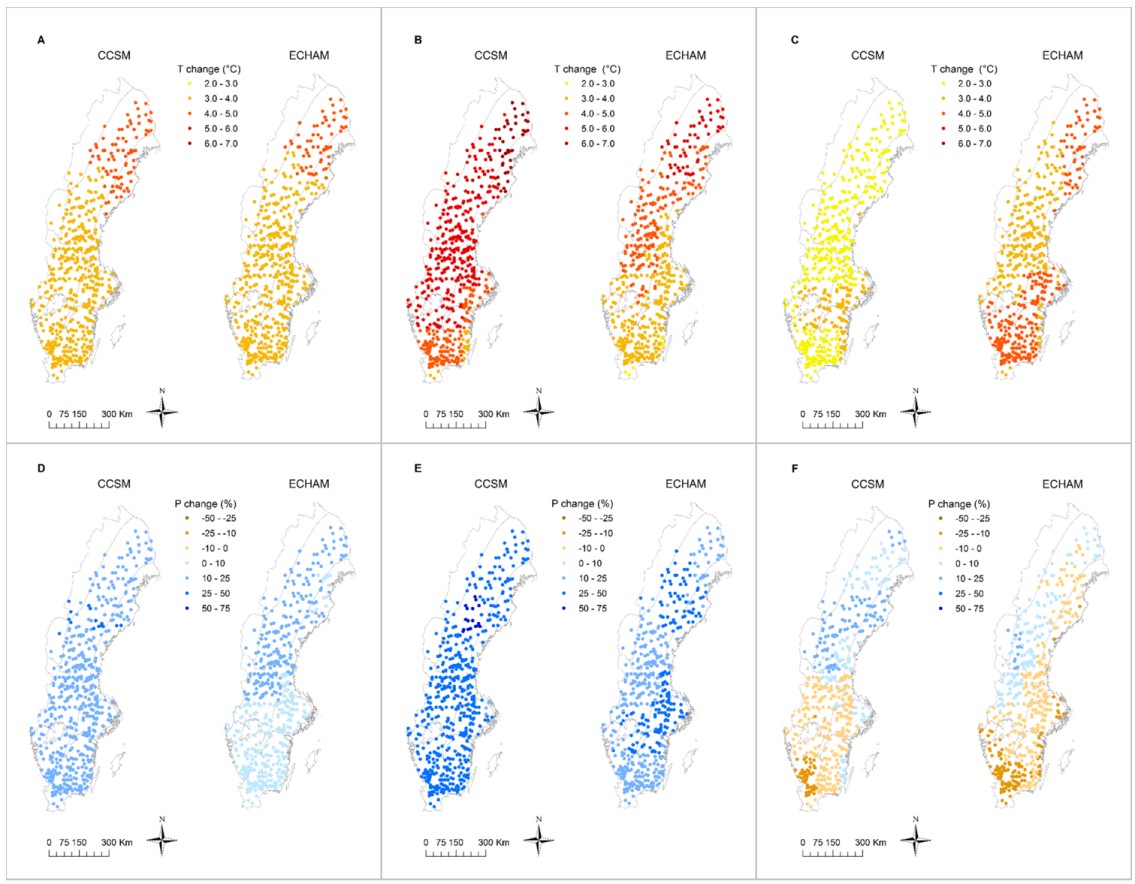

Figure 1: Differences in air temperature (T) and precipitation (P) between climate change projections by
the CCSM and ECHAM models for the period 2071-2100 as compared to a hypothetical future with no
change from the reference period of 1981-2010 (adapted from Belyazid and Zanchi, 2019). The upper
row shows: A- the difference in mean annual temperature (°C), B- mean winter temperature (°C) and C-
mean summer temperature (°C). The lower row shows the corresponding difference in precipitation (%)
annually (D), for the winter season (E) and for summer (F).

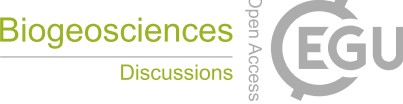



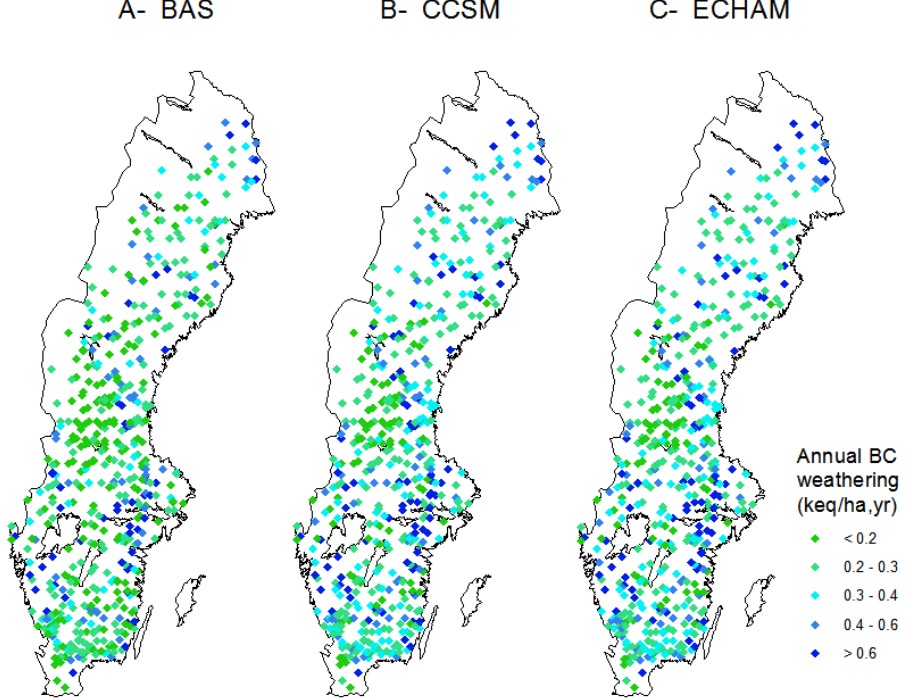

Figure 2: Geographical distributions of mineral weathering rates (keq.ha$^{-1}$.yr$^{-1}$) in the unsaturated soil at 544 productive forest sites. A- without changes in climate, B- using CCSM climate projections, and C- using ECHAM climate projections.





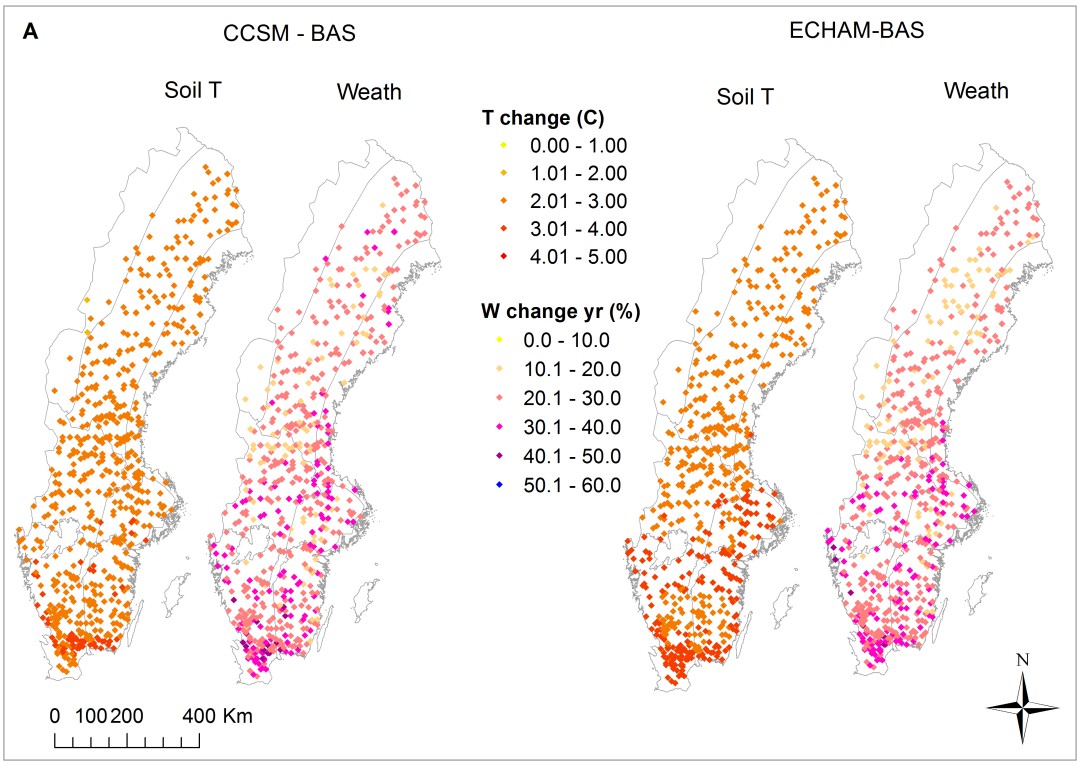

Figure 3: Geographical distribution of changes in soil temperature (°C) and weathering rates (%) between CCSM and the baseline and between ECHAM and the baseline.





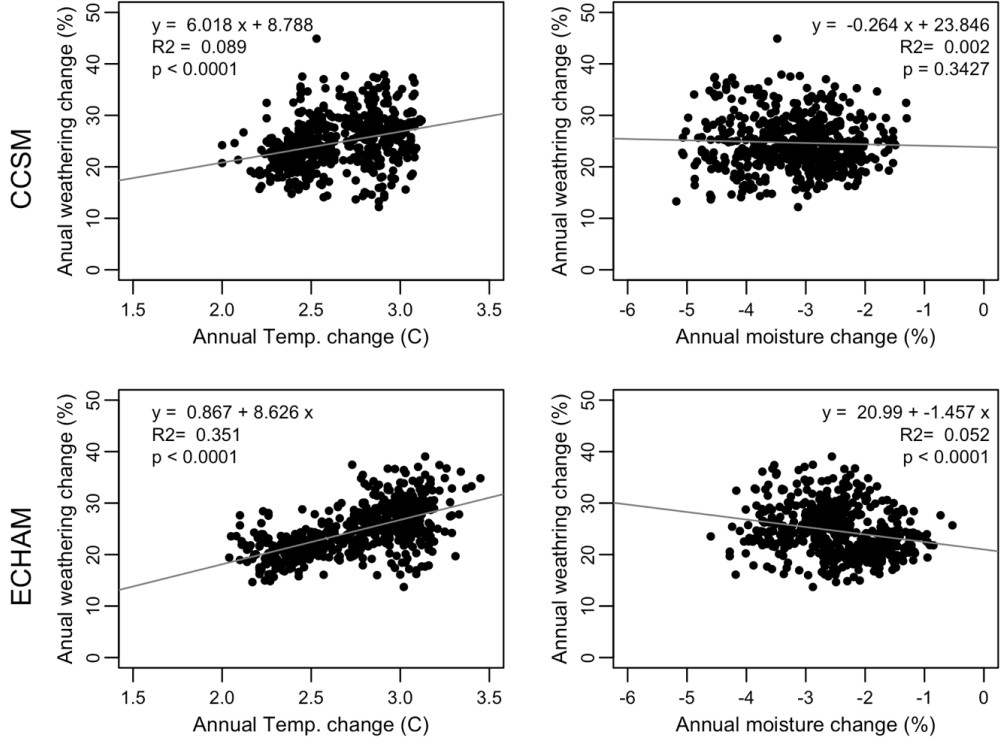

Figure 4: Linear regressions between changes in annual soil temperature and moisture and change in annual weathering rates, under two climate scenarios according to the CCSM (top row) and ECHAM models (bottom row).



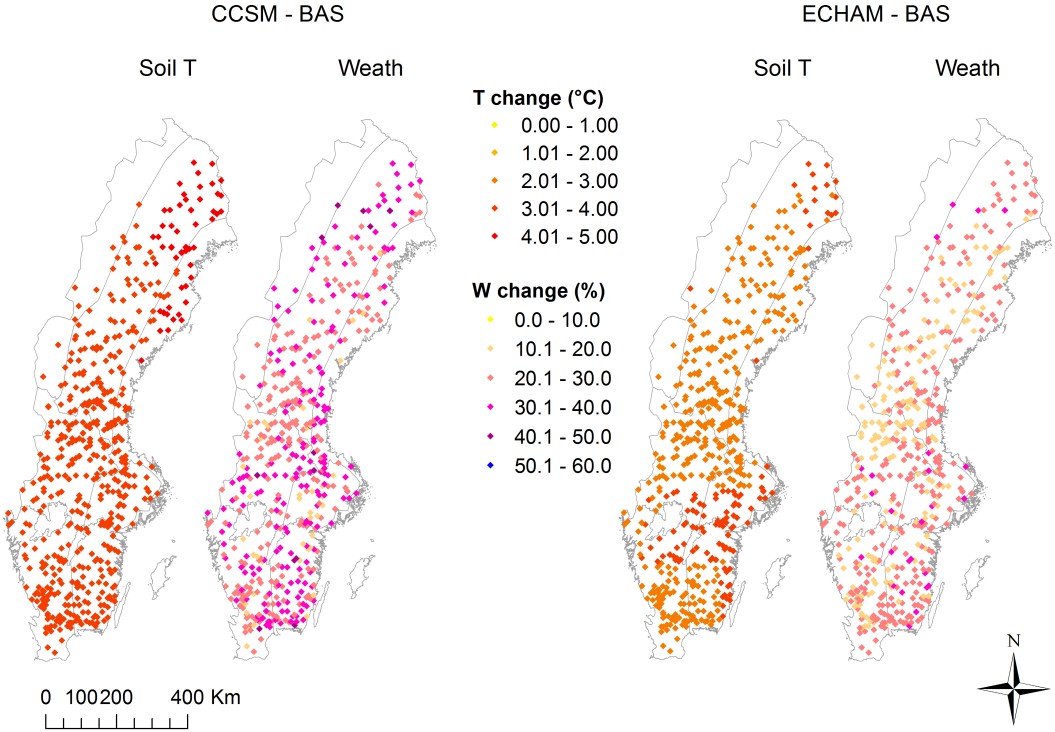

Figure 5: Differences in summer soil temperatures and weathering rates according to A- CCSM climate projections and B- ECHAM climate projections.





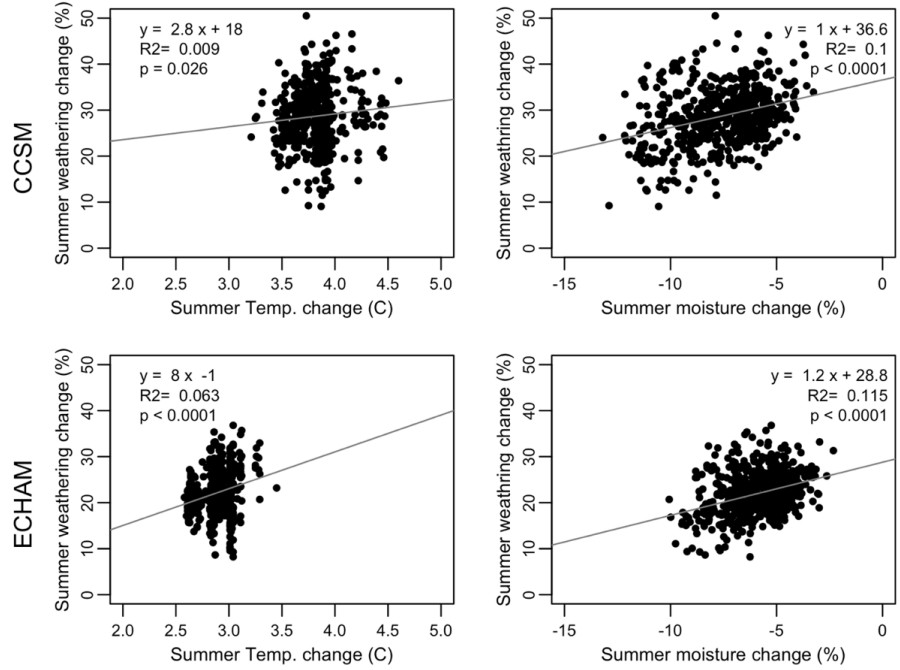

Figure 6: Correlation between the change in weathering rates and the change in soil temperature and moisture for the summer season over the period 2070-2100.





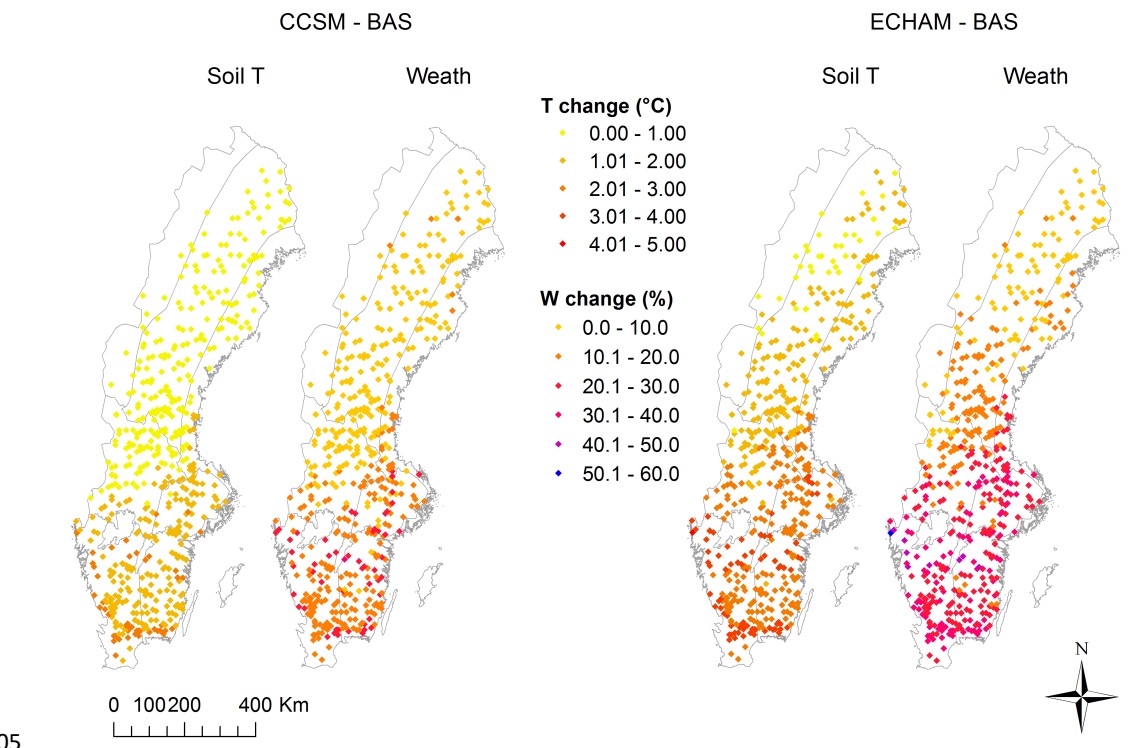

Figure 7: Changes in winter soil temperature and weathering rates under: A- CCSM climate and B- ECHAM climate.





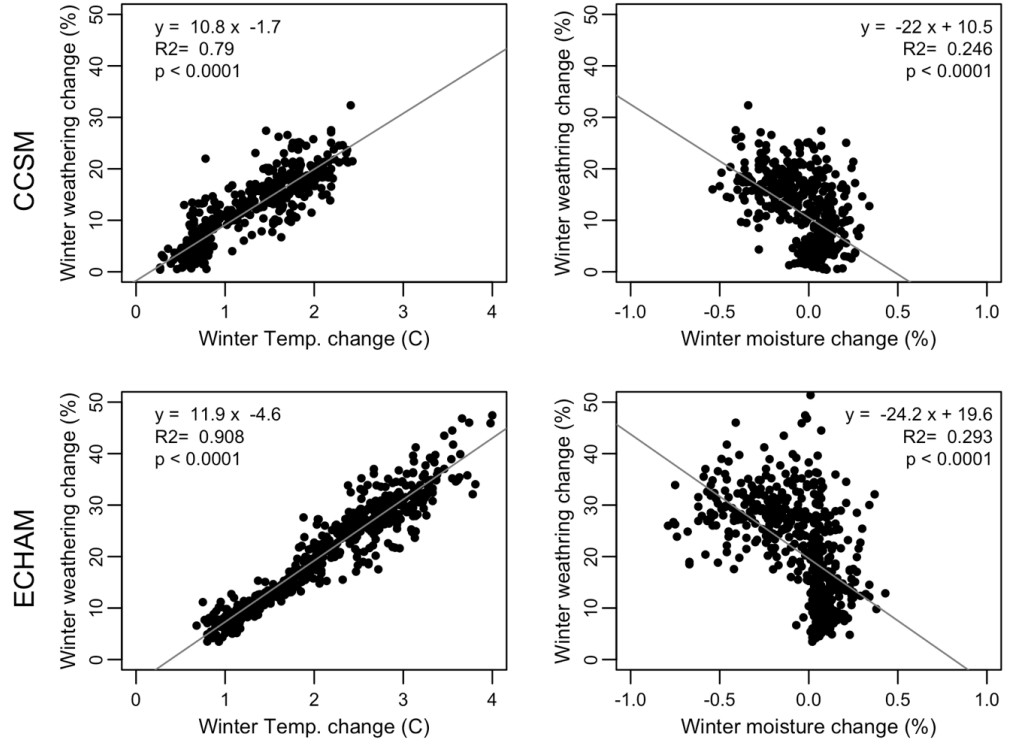

Figure 8: Correlations between the change in winter weathering rates and soil temperature and moisture under CCSM (top row) and ECHAM (bottom row) climate projections.