# Peer review of "Water limitation may restrict the positive effect of higher temperatures on weathering rates in forest soils"

_Biogeosciences, 2019_

## Referee Comment (RC1) · Anonymous Referee #1 · 28 Mar 2019

General comment:

The present manuscript describes the use of a model to calculate weathering rates under two different climatic scenarios where temperature and precipitation are expected to greatly vary within some years. The purpose of the study is very interesting since normally the expected increase of temperature is linked to an increase of the weathering rates and therefore availability of base cations to soil, which can contribute to a myriad of pedological and biological processes. However, as it was correctly pointed out by the authors, the weathering rates are dependent of other factors then temperature and soil moisture and in parallel these two parameters will be dependent on other

complex occurring processes. However the study has two main problems. Firstly, no precise information is provided concerning the model used. Authors do not specify if soil moisture is simulated along the soil profile and if the soil is divided into different layers (this is relevant since the first layers are essentially organic). Also, there is no information on the input required for this simulation and how it was obtained such as the organic matter and nutrient concentrations, the different soil types, parent material and also the different occur minerals at different forest sites, the vegetation parameters required to correctly use a vegetation growth model since different trees will have different litter rates and constitution and water uptake dynamics. Finally, authors do not specify how they have calibrate the model for these forest sites and ultimately how did they validate it. Authors should demonstrate that the model is correctly simulating the effects induced by temperature and soil moisture in these places. The second problem of the manuscript is the range of results and discussion achieved. Once authors point out that climatic changes have complex effects on weathering and that several processes are affected by the changes that ultimately can interfere with weathering it would be very interesting to understand what is governing the weathering rates or by other words, which processes are mostly being influenced by these changes and what is causing the results. As it is now the manuscript describes only the correlations between temperature and soil moisture and the weathering rates but we have no clue of what is behind these trends. When using a model, we can focus on specific flows and understand how the different pools are being affected. The manuscript tittle is a good example of the manuscript fail to provide more effective answers. When stating in the tittle that "Water limitation may restrict..." when that was already the hypothesis do not show much improvement. As a conclusion, while the study has a valid and significant goal, it fails along the day to deliver the important messages.

Detailed comments:

Line 18 – "decomposition" – Authors often use this term along the manuscript but it would be more accurate to say "organic decomposition".

Line 28 – 32 – this conclusion is general and authors have only simulated what happens in forest sites in Sweden! It is possible that climatic scenarios for different soils, vegetation covers and land uses result in completely different results. Authors should stick to their study cases.

Line 34-35 – This sentence is too radical. It was not the economic activity that caused environmental impacts, but instead the "bad management in that economic activity".

Line 45 – There is a typo here (Ref).

Line 88 – Which nutrients and which elements? At least the most important.

Line 91-92 – $CO_2$ and $H+$ are not correctly formatted. There are several typos like these along the manuscript.

Line 94 – Dissolution rate of $H+$???

Line 125 – What are the differences between the different forest sites: vegetation types? Soil types? Parent materials? Topography?

Line 136 – 140 – "Forest stand history and future management" this was used as an input? How?

Line 170 – This is the first time authors are saying that the parent materials was different and that this affected the results. The clear indication of the different parent materials involved in this study should be in M&M and the effect of these inputs on the weathering rate should be in Discussion.

Line 181-182 – This is a trivial result, since soil temperature will not mimic air temperature die to thermic soil properties. The study should focus on the interaction of the different processes on the weathering rate instead.

Line 187-189 – This is also expectable since in winter the air temperature increase impacts more the soil that can be frozen. This is the case of Sweden but it will be different for sure in other places. These type of reflections should be in Discussion.

Line 215 – The sub-type if very short. Also results are focused too much on summer and winter changes and also too many figures are devoted to these cases. That information can be summarized.

Line 240 -241 – Strange sentence.

Figure 2, 3, 5 and 7 – The colour ramp should be drastic so that we can see better when the most important changes will occur, Variations between green and blue of between different yellows are not different enough.

---

## Referee Comment (RC2) · Anonymous Referee #2 · 3 Apr 2019

General remarks:

The manuscript describes how weathering rates may change under different climate projections in future. It uses a modelling approach (ForSAFE) on 544 forest sites in Sweden. One has to assume that a huge dataset is available that is used for modelling (but there is no data repository given). In my opinion, the manuscript is in a pre-mature stage and I cannot recommend it for publication. The manuscript is presented more like a technical report and I do not see much added value for the scientific community right now. Furthermore, similar studies have been carried out using another model that finally gave quite similar results. So, the question arises: why doing this exercise

again? The used model in this study seems to be "a fully dynamic ecosystem model" – but it provides the same results.

I see the following major flaws:

- no link to a data repository or at least to a summary of the data is given (e.g. as supplementary material)

- I would have expected at least a sensitivity analysis of the main parameters used. I see nice maps – but have no idea how reliable or sensitive the projected results are

- what is the novelty of this research?

- no overview of the present state of knowledge and gap in knowledge is given. There are other models and approaches. I would like to see advantages/disadvantages of existing approaches. Other modelling approaches are more physically based, e.g. the percolation theory (see e.g., Hunt and Ghanbarian, 2016). Consequently, I would expect that the theoretical concept is much better embedded to demonstrate why now a modelling exercise using ForSAFE is necessary to be performed.

As previously mention, my recommendation is to reject this manuscript. A fully rewritten and reorganised paper can be reconsidered.

Details:

L. 45: Ref.?

L. 49-54: relation to climate?

L. 74-77: should be extended. There is not only the programme ForSAFE. What about others? Better overview of current knowledge. Derive research questions.

L. 126-144: what about a sensitivity analysis: atmospheric deposition, forest management, . . .

L. 170-171: make reference to table

L. 182: how is the increase in soil temperature calculated? Which soil depth?

L. 209: how is soil moisture modelled? Which soil depth?

L. 225-228: . . . so, why doing this exercise? There seems no difference to PROFILE which does not seem to be surprising because it is the basis of ForSAFE (see L. 89).

References:

Hunt, A. G., and Ghanbarian, B. (2016). Percolation Theory for Solute Transport in Porous Media: Geochemistry, Geomorphology, and Carbon Cycling. Water Resour. Res. 52, 7444–7459. doi: 10.1002/2016WR019289

---

## Referee Comment (RC3) · Bradley W. Goodfellow (Referee) · 4 Apr 2019

The paper by Salim Belyazid and coauthors reports the results of modelling of the effects of predicted future climate change, with respects specially to changes in soil temperatures and soil moisture, on future chemical weathering rates in forest soils. They use an 'integrated forest ecosystem model' known as ForSAFE to model chemical weathering on 544 managed forest sites distruted through Sweden. They derive their future climate predictions from two general circulation models. From their modeling, they conclude that soil moisture will decrease in summer under predicted future warming, which could limit expected future increases in weathering rates attributable

to higher temperatures.

I write this review from the perspective of someone who is not a modeler but rather is a geomorphologist with knowledge of weathering. I thought the paper was really disappointing, given that this is an interesting topic and that the paper started reasonably well. The Abstract was clear and concise, the Introduction set the problem up, and the Methods were explained. However, from there the manuscript deteriorates with some fairly uninspiring results, a discussion that reads as esoteric and largely irrelevant to the results, and no Conclusions. The authors discuss various shortcomings with weathering models but offer no interpretation of their Results. There also seems to be almost no connection between the model and the physical environment, depite using 544 forest soil sites (boreal forest, soils developed on glacial diamicts, temperate to Arctic climatic range, mineral derived nutrients, podzols, etc, are all terms reflective of the physical environment that I was looking for but which were missing). Also, the important effects of soil erosion on chemical weathering rates are completely overlooked and there is no consideration of the temporal evolution of these soils derived from glacial sediments, which may be largely unrelated to climate. For example, what happens over time as easily available mineral nutrients in the sediments are depleted? Is there any bedrock weathering and how might that change in the future? The soils on which many of these forests are located are frequently thin and clastic. What about the effects of fire under a future warmer climate on these managed forests, in terms of nutrient uptake, nutrient cycling, effects on surface runoff, and vegetation assemblages, which might also be important enough to at least warrant a mention? The thread of their narrative is difficult to follow and I missed having a Conclusion section. The paper simply finishes with this sentence: "To reduce the uncertainties connected to those regressions, a revision of the regression is required, where the newest technology can be used". It really says nothing, which in a way sums up this paper. It's a pity because I think the modelling exercise that the authors have completed has merit and it could be worthy of publication, but certainly is not, in my view, in its present form.

In addition to this general impression, I list the following specific points: I have enormous sympathy for non-native English speakers having to write scientific English. It's not easy. However, the manuscript does suffer from it being difficult to read and from numerous grammatical errors, especially through the Results and Discussion sections. A revised version would benefit from more detailed proof reading.

15: Previous estimates of weathering rates? And what about temperature controls on reaction kinetics? 20: For those from a different field, it could be worthwhile explaining 'productive'. We can guess what it probably means but it would be good to have it defined. Also, what is modeled? 25: Elevated air temperatures should appear earlier 30: This could be shortened 35: It is unclear what exceedance of planetary boundaries means, its connection with climate change, and its relevance for this paper. 40: Why is an increase in biomass production needed to meet the stated goals? 50: Grammatical error with the references. Be specific and substitute tree harvesting for forestry!? 55: Compensation potential – with respect to acidity? The reader can often guess what you mean but the writing should be much clearer. 60: Losses of what? 65: Accelerated is preferable to higher because the latter might mean larger trees. 70: Could be worth explaining radicals or being more specific with respect to listing those. Also the number and intensity of wetting and drying cycles might be important. "Net effect is positive or negative" be explicit: "on chemical weathering of soils". Also weathering encompasses physical and chemical processes, and physical processes related to freeze-thaw/frost cracking will change in a different way to chemical processes, under a warmer climate. 80: Are you explicitly modelling all of the unsaturated zone? What about in locations where it extends below the maximum rooting depth? What is the maximum rooting depth? 90: "and" organic cations. And "$H_2O$" with the 2 as subscript. 95: What are the different layers of the simulated soil? 125: Hydrology-related fluxes of what? 130: No previous mention of the listed atmospheric pollutants. Explain their importance to this study and do this earlier in the paper. 150: Explain why you use these two particular models: ECHAM5 GCM and CCSM3 GCM. Is "story line" the correct term?

RESULTS: In general, I found this section to be hard to follow. 175: Temporal or spatial 'variability'? 175-180: These two sentences read as being in conflict with each other. 190: Is it really both climate models? In the CCSM result in Figure 4 it appears that variance increases with soil temperature increase. 195: There really is no correlation between annual weathering rate and predicted moisture change. Is the correlation really significant? You are explaining 5% of the variance. The regression line very weakly indicates that annual weathering will increase with reduced soil moisture. Why would this be so and if you consider this result to be significant then why isn't is further explored in the Discussion?

DISCUSSION: There is essentially no exploration of the results but lots of esoteric model details. The sentence below 230 is essentially a repeat of 230. Part of the disparity in what? 235: Geophysical properties – what about geochemical properties? Soil coarse material – this is the first mention of grain sizes and their importance to the study needs to be introduced near the start of the paper. "Fine earth" – is that silt + clay? "Estimation method" – what is being estimated? 240: "User defined possible materials" sounds like something out of a robotic University HR department. It is incomprehensible here. Also a concluding sentence is needed for this paragraph. What is the key point for the reader? 250: It is unclear what this sentence means 265: It is entirely unclear what the concluding sentence to this paragraph means. 275: Exposed mineral surface – is this reactive surface area?

CONCLUSIONS: Where are they?

TABLES AND FIGURES Table 1: Might be better in a plot!? Define "future climate" in the footnotes. It is not a constant, but rather is evolving. 370: What is "increment of weathering"? 385: Define BC in the caption. 390: Figure 4: only the bottom left hand figure shows a correlation, and it is weak. 400: Figure 6: Is it even worth showing these? The highest R2 of the 4 plots is 0.1. 410: Figure 8. Here you have something but you make no attempt in the Discussion to explain it. This result may reflect an expected chemical weathering threshold around zero degrees. Increases in winter temperatures, such that the soil is unfrozen for longer, is likely to be important in Swedish forest soils and your results support this.

In general, with respect to the figures explain how you calculated your statistics. The equations are all expressed in terms of X and Y but the parameters vary between plots, so should be individually defined for each plot. R2 should be R2 (superscript).

―――――――――――――――――

---

## Author Comment (AC1) · 17 May 2019

Thank you very much for the accurate summary of what we wanted to achieve with this paper. We are very thankful for the valuable references you indicate, and admit that we were not aware about them. The three papers you sent are very relevant to our work and we will certainly have good use of them in the revision of the paper.

---

## Author Comment (AC2) · 17 May 2019

Anonymous Referee #1 General comment: The present manuscript describes the use of a model to calculate weathering rates under two different climatic scenarios where temperature and precipitation are expected to greatly vary within some years. The purpose of the study is very interesting since normally the expected increase of temperature is linked to an increase of the weathering rates and therefore availability of base cations to soil, which can contribute to a myriad of pedological and biological processes. However, as it was correctly pointed out by the authors, the weathering rates are dependent of other factors then temperature and soil moisture and in parallel these two parameters will be dependent on other complex occurring processes. However the study has two main problems. Firstly, no precise information is provided concerning the model used. Authors do not specify if soil moisture is simulated along the soil profile and if the soil is divided into different layers (this is relevant since the first layers are essentially organic). Also, there is no information on the input required for this simulation and how it was obtained such as the organic matter and nutrient concentrations, the different soil types, parent material and also the different occur minerals at different forest sites, the vegetation parameters required to correctly use a vegetation growth model since different trees will have different litter rates and constitution and water uptake dynamics. Finally, authors do not specify how they have calibrate the model for these forest sites and ultimately how did they validate it. Authors should demonstrate that the model is correctly simulating the effects induced by temperature and soil moisture in these places. The second problem of the manuscript is the range of results and discussion achieved. Once authors point out that climatic changes have complex effects on weathering and that several processes are affected by the changes that ultimately can interfere with weathering it would be very interesting to understand what is governing the weathering rates or by other words, which processes are mostly being influenced by these changes and what is causing the results. As it is now the manuscript describes only the correlations between temperature and soil moisture and the weathering rates but we have no clue of what is behind these trends. When using a model, we can focus on specific flows and understand how the different pools are being affected. The manuscript tittle is a good example of the manuscript fail to provide more effective answers. When stating in the tittle that "Water limitation may restrict. . ." when that was already the hypothesis do not show much improvement. As a conclusion, while the study has a valid and significant goal, it fails along the day to deliver the important messages.

Response to Reviewer 1's general comments: The summary and shortcomings pointed out are precise and very relevant. We are thankful for the constructive criticism and based on that will modify the manuscript in two ways: 1- we will give a more comprehensive description of the model and the datasets used, and 2- we will expand the results and discussion to explain the processes and pathways through which climate is affecting weathering in the simulations. We would like to add a section where we present and discuss the responses of the four weathering pathways to the changes in soil temperature and moisture, and how these changes affect the dissolution rates (through action of the kinetics or through affecting solution saturation levels). We will also reformulate the title to better represent the core message of the paper.

Detailed comments: Line 18 – "decomposition" – Authors often use this term along the manuscript but it would be more accurate to say "organic decomposition". We will make the use of the term "decomposition" more explicit to avoid any misrepresentation.

Line 28 – 32 – this conclusion is general and authors have only simulated what happens in forest sites in Sweden! It is possible that climatic scenarios for different soils, vegetation covers and land uses result in completely different results. Authors should stick to their study cases. Thank you for pointing this out. We will reformulate the sentences to relate to the study area.

Line 34-35 – This sentence is too radical. It was not the economic activity that caused environmental impacts, but instead the "bad management in that economic activity". The reviewer's view is correct, we will rephrase.

Line 45 – There is a typo here (Ref). A relevant reference will be included here.

Line 88 – Which nutrients and which elements? At least the most important. We will explicitly spell out the nutrients and other elements simulated in ForSAFE.

Line 91-92 – $CO_2$ and $H^+$ are not correctly formatted. There are several typos like these along the manuscript. Thank you for pointing this out, we will review these throughout the document.

Line 94 – Dissolution rate of $H^+$??? We meant to refer to the dissolution pathways controlled by $H^+$. We will reformulate this sentence.

Line 125 – What are the differences between the different forest sites: vegetation types? Soil types? Parent materials? Topography? The simulated forest stands are all productive, managed forests, with predominantly spruce and pine forests. The biggest majority are on podzols, but most often than not unrelated to the bedrock material (due to till movements during glaciation). All sites are on well drained soils, usually with very gentle slopes. This information will be included in the paper with relevant references.

Line 136 – 140 – "Forest stand history and future management" this was used as an input? How? Forest management (including history) tells us when a stand was planted, thinned or cut. The model uses this information to reproduce the same interventions as on the field (planting, thinning and final felling).

Line 170 – This is the first time authors are saying that the parent materials was different and that this affected the results. The clear indication of the different parent materials involved in this study should be in M&M and the effect of these inputs on the weathering rate should be in Discussion. Yes we agree with the reviewer, we will include a description of the parent material in the methods.

Line 181-182 – This is a trivial result, since soil temperature will not mimic air temperature die to thermic soil properties. The study should focus on the interaction of the different processes on the weathering rate instead. We agree with the reviewer. We will still keep this statement to stress the importance of properly assessing soil temperature, but will put more focus on the response of the different weathering control processes to increased temperature (this is also in line with critique from reviewer 2).

Line 187-189 – This is also expectable since in winter the air temperature increase impacts more the soil that can be frozen. This is the case of Sweden but it will be different for sure in other places. These type of reflections should be in Discussion. Yes this is possible, but in case of insulation by snow the impact in winter can be very small (which is what we show). We will point this out more clearly, and come back to it in the discussion.

Line 215 – The sub-type if very short. Also results are focused too much on summer and winter changes and also too many figures are devoted to these cases. That information can be summarized. We wanted to stress the winter-summer difference, as weathering will react different to higher temperatures under these two seasons. In summer, when soil temperature increases more in unison with air temperature, it will be soil moisture that limits the effect on weathering. While in winter, soil moisture is not limiting, but soil temperature does not increase as much. We will seriously try to summarise these into fewer maps as requested by the reviewer, while trying not to lose the message above.

Line 240 -241 – Strange sentence. Yes we understand the reviewer's reaction. We will change or remove this sentence. Figure 2, 3, 5 and 7 – The colour ramp should be drastic so that we can see better when the most important changes will occur, Variations between green and blue of between different yellows are not different enough. Thank you for pointing this out, it becomes a little difficult to see the maps objectively after making a few dozens of them. We will revise the colour scales to make the difference more visible on the maps.

---

## Author Comment (AC3) · 17 May 2019

General remarks: The manuscript describes how weathering rates may change under different climate projections in future. It uses a modelling approach (ForSAFE) on 544 forest sites in Sweden. One has to assume that a huge dataset is available that is used for modelling (but there is no data repository given). In my opinion, the manuscript is in a pre-mature stage and I cannot recommend it for publication. The manuscript is presented more like a technical report and I do not see much added value for the scientific community right now. Furthermore, similar studies have been carried out

using another model that finally gave quite similar results. So, the question arises: why doing this exercise again? The used model in this study seems to be "a fully dynamic ecosystem model" – but it provides the same results.

Response to the reviewer's general comments: This is a valuable comment as it questions the very purpose of the paper. Assuming that the reviewer refers to the earlier study by Akselsson et al (2016) which uses PROFILE to calculate weathering rates, our aim was to either confirm or question the conclusions of that study by explicitly taking into account seasonality and ecosystem feedbacks (particularly between soil and biota). In this respect, we hoped to move forward with the estimates as PROFILE is heavily dependent on user defined assumptions for uptake, litterfall and mineralisation, and soil moisture content and hydrological transport. Also, being a steady state model, it does not consider temporal dynamics. ForSAFE addresses these issues by internally calculating growth, uptake, litterfall, mineralisation and how these are controlled by soil chemistry and hydrology. It also explicitly simulate hydrology and heat transfer, allowing us to simulate the diffusion of air temperature into the soil, and responding to seasonal variations in precipitation, vapor pressure difference and evapotranspiration. That the results of this study agree with those in the work by Akselsson et al. is welcome, but the extent of the climate impact was shown here to be lower than expected earlier. Present weathering rates estimated by ForSAFE agree very well with those from PROFILE. Yet the estimated impact of future climate calculated by the two models is different for the different climate scenarios (+23% vs. +33% and +26% vs 20% from ForSAFE and PROFILE, using CCSM and ECHAM respectively). The two models simulated different periods, but when looking at the increase of weathering per degree increase in air temperature, ForSAFE give a significantly lower estimate. What we want to point out is that considering ecosystem dynamics, together with seasonality, can significantly impact our estimates of the effects of climate change on mineral weathering. The reviewer's critical response is very valuable for making us aware that this message maybe did not come through. We would like to reorganise the paper in two ways to bring the paper up to the requested level: 1- a more comprehensive

section for materials and methods, including an evaluation of the model, and 2- an overhaul of the results and discussion sections (also in accordance with reviewer 1) to present more dynamic process results and lift more the difference between this study and previous work.

I see the following major flaws: - no link to a data repository or at least to a summary of the data is given (e.g. as supplementary material) We will revise this and provide a detailed summary of the data used, rather than referring to other work.

- I would have expected at least a sensitivity analysis of the main parameters used. I see nice maps – but have no idea how reliable or sensitive the projected results are We are uncertain if a sensitivity analysis is really imperative here. Unlike PROFILE, we do not need to assume things like soil moisture in ForSAFE. Sensitivity analyses of specific parameters (eg. controlling hydrology) have been done in other studies (eg. Zanchi et al., 2016; Kronnäs et al., this issue). We will however include a clear assessment of model output against available empirical data.

- what is the novelty of this research? The novelty of the paper is in considering internal ecosystem feedbacks and high temporal resolution is forecasting the expected gain in weathering from climate change. This, in comparison to a steady state model (eg. PROFILE in Akselsson et al., 2016), limits the increase in weathering per degree of temperature from around 10% to below 8%. Again, thank you for making us aware that we failed to communicate the this.

- no overview of the present state of knowledge and gap in knowledge is given. There are other models and approaches. I would like to see advantages/disadvantages of existing approaches. Other modelling approaches are more physically based, e.g. the percolation theory (see e.g., Hunt and Ghanbarian, 2016). Consequently, I would expect that the theoretical concept is much better embedded to demonstrate why now a modelling exercise using ForSAFE is necessary to be performed. The approach in this study is indeed different from that in Hunt and Ghanbarjan. We do not really

understand what the reviewer aims at in the last part this comment. The first part however would certainly give more context to the paper, and we will include a short but up to date background section on existing approaches. The purpose of this paper not being an assessment of approaches, we do not clearly see the need to motivate the choice of model more than the requirement for process and temporal dynamics on a decadal scale.

As previously mention, my recommendation is to reject this manuscript. A fully rewritten and reorganised paper can be reconsidered. We would like to revise the paper as described in the responses.

Details: L. 45: Ref.? Thank for pointing out the omission. We will refer to the relevant source.

L. 49-54: relation to climate? Iwald et al. showed no relation to climate. The study is referred to here to show that base cation budgets are very tight, reducing the error margins we can accept in estimates and requiring as precis weathering estimates as we can get.

L. 74-77: should be extended. There is not only the programme ForSAFE. What about others? Better overview of current knowledge. Derive research questions. Yes we agree with the reviewer, we will include a more extensive review of other methods. We will also try to formulate specific research questions that will lead to the aim.

L. 126-144: what about a sensitivity analysis: atmospheric deposition, forest management, . . . A separate study by Kronnäs et al looked more at these aspects and how ForSAFE responds to them. We do not necessarily see the need for these here, as we try to focus on climate.

L. 170-171: make reference to table We will refer to the table earlier in this paragraph.

L. 182: how is the increase in soil temperature calculated? Which soil depth? L. 209: how is soil moisture modelled? Which soil depth? We realise these are aspects we

did not describe. We will add a section in the methods describing the modelled soil depths, and how the model deals with soil moisture and temperature. Because the model simulates three to five layers at each modelled stand, we show the cumulative weathering rate for the entire profile, but only soil temperature at the middle of the profile (i.e. at half the simulated depth, which usually is between 40cm and 50cm depending on the site).

L. 225-228: . . . so, why doing this exercise? There seems no difference to PROFILE which does not seem to be surprising because it is the basis of ForSAFE (see L. 89). The mineral dissolution equations are the same between PROFILE and ForSAFE. The difference is that PROFILE takes all other fluxes to and from the soil solution as inputs (uptake, mineralisation, water percolation. . .) and considers no cation exchange. ForSAFE takes only climate, atmospheric deposition and forest management, while all other processes are modelled internally. PROFILE produces steady state weathering levels, while ForSAFE allows us to follow the change over time, with sometimes significant delays in response as the ecosystem adapts to new conditions. Most importantly, and the reason why we needed this study, is that ForSAFE can account on its own for the feedback between plants and soil. As it gets warmer, more water will be lost because of evapotranspiration, and this effect is even more expressed during the growing season. Akselsson et al shows present weathering rates at selected locations, while here we try to capture current (and double check them with the literature) and future weathering rates, and we believe a tool like ForSAFE is suited for that.

References: Hunt, A. G., and Ghanbarian, B. (2016). Percolation Theory for Solute Transport in Porous Media: Geochemistry, Geomorphology, and Carbon Cycling. Water Resour. Res. 52, 7444–7459. doi: 10.1002/2016WR019289

---

## Author Comment (AC4) · 17 May 2019

Bradley W. Goodfellow (Referee) bradley.goodfellow@sgu.se

 The paper by Salim Belyazid and coauthors reports the results of modelling of the effects of predicted future climate change, with respects specially to changes in soil temperatures and soil moisture, on future chemical weathering rates in forest soils. They use an 'integrated forest ecosystem model' known as ForSAFE to model chemical weathering on 544 managed forest sites distruted through Sweden. They derive their future climate predictions from two general circulation models. From their modeling, they conclude that soil moisture will decrease

in summer under predicted future warming, which could limit expected future increases in weathering rates attributable to higher temperatures. I write this review from the perspective of someone who is not a modeler but rather is a geomorphologist with knowledge of weathering. I thought the paper was really disappointing, given that this is an interesting topic and that the paper started reasonably well. The Abstract was clear and concise, the Introduction set the problem up, and the Methods were explained. However, from there the manuscript deteriorates with some fairly uninspiring results, a discussion that reads as esoteric and largely irrelevant to the results, and no Conclusions. The authors discuss various shortcomings with weathering models but offer no interpretation of their Results. There also seems to be almost no connection between the model and the physical environment, depite using 544 forest soil sites (boreal forest, soils developed on glacial diamicts, temperate to Arctic climatic range, mineral derived nutrients, podzols, etc, are all terms reflective of the physical environment that I was looking for but which were missing). Also, the important effects of soil erosion on chemical weathering rates are completely overlooked and there is no consideration of the temporal evolution of these soils derived from glacial sediments, which may be largely unrelated to climate. For example, what happens over time as easily available mineral nutrients in the sediments are depleted? Is there any bedrock weathering and how might that change in the future? The soils on which many of these forests are located are frequently thin and clastic. What about the effects of fire under a future warmer climate on these managed forests, in terms of nutrient uptake, nutrient cycling, effects on surface runoff, and vegetation assemblages, which might also be important enough to at least warrant a mention? The thread of their narrative is difficult to follow and I missed having a Conclusion section. The paper simply finishes with this sentence: "To reduce the uncertainties connected to those regressions, a revision of the regression is required, where the newest technology can be used". It really says nothing, which in a way sums up this paper. It's a pity because I think the modelling exercise that the authors have completed has merit and it could be worthy of publication, but certainly is not, in my view, in its present form.

Response to the reviewer's general comments: We appreciate all the aspects taken up, but we want to focus the paper on the effects of climate in the coming 80 years or so. Under this time period, we do not expect any significant physical changes, and want instead to focus on the interplay of moisture and temperature, to see if the 10% increase in weathering per 1 degree increase in temperature will materialise. However, we agree that the paper will bebenfit from a more thorough description of the results and more in depth discussion (this is in line with the message from reviewers 1 and 2).

15: Previous estimates of weathering rates? And what about temperature controls on reaction kinetics? 20: For those from a different field, it could be worthwhile explaining 'productive'. We can guess what it probably means but it would be good to have it defined. Also, what is modeled? 25: Elevated air temperatures should appear earlier 30: This could be shortened 35: It is unclear what exceedance of planetary boundaries means, its connection with climate change, and its relevance for this paper. 40: Why is an increase in biomass production needed to meet the stated goals? 50: Grammatical error with the references. Be specific and substitute tree harvesting for forestry!? 55: Compensation potential – with respect to acidity? The reader can often guess what you mean but the writing should be much clearer. 60: Losses of what? 65: Accelerated is preferable to higher because the latter might mean larger trees. 70: Could be worth explaining radicals or being more specific with respect to listing those. Also the number and intensity of wetting and drying cycles might be important. "Net effect is positive or negative" be explicit: "on chemical weathering of soils". Also weathering encompasses physical and chemical processes, and physical processes related to freeze-thaw/frost cracking will change in a different way to chemical processes, under a warmer climate. 80: Are you explicitly modelling all of the unsaturated zone? What about in locations where it extends below the maximum rooting depth? What is the maximum rooting depth? 90: "and" organic cations. And "$H_2O$" with the 2 as subscript. 95: What are the different layers of the simulated soil? 125: Hydrology-related fluxes of what? 130: No previous mention of the listed atmospheric pollutants. Explain their importance to this study and do this earlier in the paper. 150: Explain why you use these two particular

models: ECHAM5 GCM and CCSM3 GCM. Is "story line" the correct term?

Thank you for the thorough comments. We will try to address them as we revise the text where we understand the request and where we see its relevance (we don't for example understand the question about planetary boundaries, productive forests or losses in line 60). Other instances, such as the request for clarification about if we model the unsaturated zone, will be addressed. Note to editors: we suspect that there is a mismatch in the line numbers referred to by Dr. Goodfellow and the version available from the journal. But we will try to look for the text indicated by the reviewer and not the line numbers.

RESULTS: In general, I found this section to be hard to follow. 175: Temporal or spatial 'variability'? 175-180: These two sentences read as being in conflict with each other. 190: Is it really both climate models? In the CCSM result in Figure 4 it appears that variance increases with soil temperature increase. 195: There really is no correlation between annual weathering rate and predicted moisture change. Is the correlation really significant? You are explaining 5% of the variance. The regression line very weakly indicates that annual weathering will increase with reduced soil moisture. Why would this be so and if you consider this result to be significant then why isn't is further explored in the Discussion? The text in line 175 refers to table 1, where the data is described as annual average for two periods. The variability is between the sites, so spatial. We will try to make this more explicit in the text. 175-180: sentence 1: there is a large variability in weathering among the sites, sentence 2: weathering will increase consistently at all sites due to climate change. The two sentences refer to different things (1- what is the weathering rate vs. 2- will it increase with climate change?). We struggle to see how these two are in conflict. 195: we agree, the correlation is insignificant, we do not understand this comment and do not disagree with the reviewer even if our formulation is not the same. Yet, as there is no correlation, why should we try to explain the very weak, insignificant trend? We do not think this is significant, as we have a statistical test telling us it is not.

DISCUSSION: There is essentially no exploration of the results but lots of esoteric model details. The sentence below 230 is essentially a repeat of 230. Part of the disparity in what? 235: Geophysical properties – what about geochemical properties? Soil coarse material – this is the first mention of grain sizes and their importance to the study needs to be introduced near the start of the paper. "Fine earth" – is that silt + clay? "Estimation method" – what is being estimated? 240: "User defined possible materials" sounds like something out of a robotic University HR department. It is incomprehensible here. Also a concluding sentence is needed for this paragraph. What is the key point for the reader? 250: It is unclear what this sentence means 265: It is entirely unclear what the concluding sentence to this paragraph means. 275: Exposed mineral surface – is this reactive surface area? We will restructure and rewrite the discussion and conclusions, as also requested by other reviewers.

CONCLUSIONS: Where are they?

TABLES AND FIGURES Table 1: Might be better in a plot!? Define "future climate" in the footnotes. It is not a constant, but rather is evolving. 370: What is "increment of weathering"? 385: Define BC in the caption. 390: Figure 4: only the bottom left hand figure shows a correlation, and it is weak. 400: Figure 6: Is it even worth showing these? The highest R2 of the 4 plots is 0.1. 410: Figure 8. Here you have something but you make no attempt in the Discussion to explain it. This result may reflect an expected chemical weathering threshold around zero degrees. Increases in winter temperatures, such that the soil is unfrozen for longer, is likely to be important in Swedish forest soils and your results support this. In general, with respect to the figures explain how you calculated your statistics. The equations are all expressed in terms of X and Y but the parameters vary between plots, so should be individually defined for each plot. R2 should be $R^2$ (superscript). We will take these comments into account as we revise the results and discussion, and include a conclusion.